# How and when doula support increases confidence in women experiencing socioeconomic adversity: Findings from a realist evaluation of an Australian volunteer doula program

**Kerryn O'Rourke**[1]*, **Jane Yelland**[2,3], **Michelle Newton**[1,4], **Touran Shafiei**[1]

**1** Judith Lumley Centre, La Trobe University, Bundoora, Victoria, Australia, **2** Murdoch Children's Research Institute, Parkville, Victoria, Australia, **3** Department General Practice, University of Melbourne, Parkville, Australia, **4** School of Nursing and Midwifery, La Trobe University, Bundoora, Victoria, Australia

* Orourke.km@students.latrobe.edu.au

**Data Availability Statement:** The sources of data reported in this study cannot be shared openly out of respect for the privacy of participants stipulated

## Abstract

How women are cared for while having a baby can have lasting effects on their lives. Women value relational care with continuity—when caregivers get to know them as individuals. Despite evidence of benefit and global policy support, few maternity care systems across the world routinely offer relational continuity. Women experiencing socioeconomic adversity have least access to good quality maternity care. Community-based doula support programs offer complementary care for these women and are known to, on average, have positive outcomes. Less understood is how, when, and why these programs work. A realist evaluation of an Australian volunteer doula program explored these questions. The program provides free social, emotional, and practical support by trained doulas during pregnancy, birth, and early parenting. This paper reports the testing and refinement of one program theory from the larger study. The theory, previously developed from key informant interviews and rapid realist review of literature, hypothesised that support increased a woman's confidence via two possible pathways—by being with her and enabling her to see her own strength and value; and by praising her, and her feeling validated as a mother. This study aimed to test the theory in realist interviews with clients, focus groups with doulas, and with routinely collected pre-post data. Seven English-speaking and six Arabic-speaking clients were interviewed, and two focus groups with a total of eight doulas were conducted, in January-February 2020. Qualitative data were analysed in relation to the hypothesised program theory. Quantitative data were analysed for differential outcomes. Formal theories of Recognition and Relational reflexivity supported explanatory understanding. The refined program theory, *Recognition*, explains how and when a doula's recognition of a woman, increases confidence, or not. Five context-mechanism-outcome configurations lead to five outcomes that differ by nature and longevity, including absence of felt confidence.

in the Human Research and Ethics Committee's (HREC) approval. Data requests should be made to La Trobe University HREC (study HEC19388) at humanethics@latrobe.edu.au.

**Funding:** Birth for Humankind provided KO with a PhD scholarship and project funding for engagement of bicultural researcher. https://birthforhumankind.org/ The funders had no role in study design, data collection and analysis, decision to publish, or preparation of the manuscript.

**Competing interests:** The authors have declared that no competing interests exist.

## Introduction

The time of pregnancy and having a new baby is often one of great significance in a woman's life course. The way women are cared for at this time can have profound and lasting effects on their health and wellbeing. It is well established that the best maternity care outcomes are achieved by relational care with continuity—provided by the same caregiver over time [1, 2]. Despite strong evidence of benefit and global policy support [3, 4], models that offer relational continuity are not routinely available in most mainstream maternity care systems across the world. Most women receive care likened to 'factory lines' of impersonal, rushed and standardised care [5–7]. This is particularly detrimental to women living with socioeconomic disadvantage, who more often experience their maternity care as poor, distressing or traumatic due to factors such as insufficient information, lack of kindness and respect, insensitivity, and discrimination [8–11].

In response to these inequities, community-based volunteer doula programs have emerged in numerous Western countries over the past decade, providing non-medical, continuous social, emotional, and practical support to women experiencing socioeconomic adversity. There is abundant evidence of doula support programs improving experiences of care and outcomes for women from disadvantaged sociocultural backgrounds [eg. 12–20]. Less understood is how, when, and why these programs work. Doula support offers relational continuity, but detail about relational mechanisms, how these are affected by context, and their various and differential outcomes, are not well understood.

These questions form the basis of the realist evaluation of a community-based volunteer doula support program in Melbourne, Australia. The program, funded by philanthropy and operating since 2014, is the only of its kind in Australia. Volunteer doulas provide free support to women experiencing financial hardship and one or more other indicators of social adversity, such as refugee background, homelessness, or complex trauma. The volunteers come from a range of backgrounds including as private doulas, midwives and midwifery students, and bicultural workers. They typically provide two to three visits to women during pregnancy, support during labour and birth, and two postnatal visits [21]. The program has not previously been evaluated.

Realist evaluation is a theory driven logic of inquiry that assumes programs 'work', (have successful outcomes) only insofar as they introduce the appropriate resources and prompt reasoning (mechanisms) in groups of people in the appropriate conditions (contexts) [22]. A realist evaluation seeks to find *what works*, *when*, *how*, *for whom and in what circumstances*. It starts with the development of program theories, structured in accordance with this realist understanding—as *context-mechanism-outcome* (CMO) configurations. Mechanisms are further conceptualised as *resources* provided by a program, and human *reasoning* in response. Data are then collected to test and refine the theories. Resulting refined theories provide plausible explanations for the workings of a program [22].

For stage 1 of this study, program theories were developed from realist interviews with key informants who designed or were working in the program, followed by a rapid realist review of literature. Seven resulting hypothesised theories covered contexts and mechanisms leading to outcomes at the program implementation level (such as having the right doulas in the program), outcomes for women (such as increased confidence) and outcomes in the maternity care system (such as increased accountability of professional care providers). The theories and their development are published elsewhere [23].

Four theories were prioritised for the evaluation. This paper reports the testing and refinement of one of these theories, called *Being by her side*. Comprised of two CMO configurations, the theory hypothesised how, when and for whom doula support causes a woman (client) to

feel dignified and confident. The first CMO suggested that when a woman is experiencing socioeconomic disadvantage *(context)* and is well matched with her doula *(context)*, the doula is 'by her side', 'gets' her, gives time, attention and respect, and believes in her *(resource)*. In response, the woman sees her own strength and value *(reasoning)*, and this leads to feeling dignified and confident *(outcome)*. The second CMO hypothesised that when a doula is kind and non-judgemental *(context)*, she notices a woman's small wins, and praises the woman as a mother *(resource)*. In response, the woman feels validated as a mother *(reasoning)* and then dignified and confident *(outcome)*. This study aims to test and refine the theory, to inform program refinement.

## Methods

### Evaluation design

The methods were iterative, with each stage informing the next. Fig 1 outlines the design, highlighting stages 2 and 3 as within scope of this paper.

### Data collection

Mixed methods were used to accumulate knowledge on different parts of the context-mechanism-outcome configurations (CMOs) of theories developed in stage 1 [22, 24]. Elements of the CMOs were put to women (clients) in realist interviews, and doulas in focus groups, for comment, with a view to confirming, denying and refining them [22, 25]. Quantitative data collected by Birth for Humankind from women about their birth confidence before and after doula support were extracted for analysis, to further understanding of program outcomes and outcome patterns.

**Client interviews.** Women's cultural backgrounds were considered program contexts to test, so sampling aimed for culturally diversity. Women from the two largest client language groups in the program (English and Arabic) were included, as the project did not have the resources to work with more language groups. English speaking women were those who had indicated at program intake that English was their primary language, as well as those with other primary languages but indicated no need for an interpreter to communicate in English.

Women were otherwise eligible if they were aged 16 or over and had their babies three to nine months prior to the interview period (to minimise birth's possible euphoric 'halo effect' and maximise recall) [26, 27]. Women were excluded if known to have a serious mental illness, or seriously ill or deceased baby, to not increase distress. Women who identified as Aboriginal or Torres Strait Islander, and those who would require consent from a guardian were excluded due to low reported numbers and their participation requiring methods not within the study scope.

English-speaking women whose babies were aged between three and nine months were immediately eligible to be interviewed and were invited by text message from Birth for Humankind in early December 2019. Invitations were sent weekly to newly eligible women (as their baby reached three months of age), until mid-January 2020. An initial message and two weekly reminders were sent, requesting that interested women contact the researchers. Eligible Arabic speaking women were invited by text message (in Arabic) in January-February 2020 and followed up by telephone call from a bicultural community worker contracted by Birth for Humankind. Different cultural positionalities and languages spoken between the Anglo-Western lead researcher [KO] and prospective Arabic speaking participants were addressed by collaborating with a bicultural researcher [28]. The design and facilitation of interview methods for Arabic speaking women are detailed further in a separate paper [29]. Participant information and consent processes for English speaking and Arabic speaking participants were

**1. Theory development**

Key informant interviews

Rapid realist review of literature

**2. Data collection**

Realist interviews with clients
- English speaking women
- Arabic speaking women

Focus groups with doulas

Extraction of routinely collected quantitative data

**3-4. Theory testing & refinement**

Analysis of all data with incorporation of formal theory.

**Fig 1. Evaluation design, with scope of this paper highlighted.**

provided by telephone, email or in person before interviews, in accordance with women's preferences and literacy levels.

The interview schedule was developed from hypothesised theories. It was piloted in English with research colleagues, and a past doula program client not eligible to participate due to the (older) age of her baby. The bicultural researcher reviewed the schedule for cultural appropriateness and translated it into Modern Standard Arabic [30–32]. Three pilot interviews were conducted with Arabic speaking new mothers who had no association with the doula program. The piloting informed minor adjustments to the interview questions and process, to ensure

cultural acceptability, safety, and feasibility of co-facilitation—led in English and interpreted [33–35].

Interviews with English speakers were conducted by KO, by telephone (at clients' preference) in January 2020. Interviews with Arabic speakers were co-facilitated by KO and the bicultural researcher in February 2020, by telephone or face-to-face in private rooms of community centres that were easily accessible by the participants. The interviews were facilitated like conversations, with a realist "teacher-learner" [22] approach. Participants were taught bits of theory (framed neutrally), and in response, the participants taught the interviewer/s how the theory did or did not work for them [36–38]. For example, the hypothesised mechanism-outcome configuration of the doula giving time, attention and respect, and believing in the woman *(resource)*, and the woman seeing her own strength and value *(reasoning)*, leading her to feeling dignified and confident *(outcome)*, was tested with the questions, "*Can you tell me what kinds of support your doula gave you*?, *When she was with you (at home, in hospital or elsewhere. . .) what was she doing*? *And how did you feel*? *Was there something she did or said that really stands out in your memory*? *If yes, can you tell me the story*? These questions did not assume a particular kind of experience/reasoning or outcome. The interview schedule was adapted overtime in response to theory refinements. For example, the outcome of increased confidence was not initially explored in relation to time, but this became important after some participants shared that their increased confidence had lasted beyond the support. Participants were thanked with gifts at the end of the interview.

All interviews were audio-recorded, and the interviewers took field notes and debriefed. The interviews were professionally transcribed, and transcriptions were checked against the audio files. Interview data were considered co-constructions of explanations. Data from interviews co-facilitated with the bicultural researcher were understood as three-way co-constructions [39, 40].

**Doula focus groups.** The focus group schedule was developed from hypothesised program theories and piloted with five research colleagues. All current and past doulas were eligible and invited to participate in focus groups. Invitations by email and text message were sent from Birth for Humankind. The initial invitation and two weekly reminders were sent in December 2019—January 2020, requesting interested doulas to contact the researchers. Participant information was sent by email. Doulas consented and gave information about their availability via email.

Two focus groups, each of two hours duration were conducted in February 2020, after business hours at the Birth for Humankind office, when program staff and management were not present. The groups were facilitated by KO and JY in English, as all participants were comfortable to participate in English. The realist teacher-learner approach was applied (as previously described). The focus groups were audio-recorded, and the facilitators debriefed and took field notes. The focus groups were professionally transcribed, and transcriptions were checked against the audio files.

**Routinely collected data.** Data collected by Birth for Humankind were extracted, deidentified and provided to the researchers in September 2021. The dataset contained records of all women referred to the program since its inception in 2014. Included was referral information (i.e. socio-demographic), intake information including assessment of women's birth confidence (see next paragraph), birth information, and three-to-six-month follow-up retrospective assessment including women's birth confidence as felt at the time of giving birth. Some items had been collected since 2014, while others, such as the pre-post measures of birth confidence, were added in 2019.

The pre and post support birth confidence measures used in this study were women's answers to two questions asked by telephone, using interpreters where required. The first was

at the time of program intake by a program coordinator *(How confident do you feel about giving birth?)*, and then followed up at three to six months after birth *(How confident did you feel about your birth?)*, by a program coordinator, contracted bicultural worker or administration assistant. Both assessments were made on a five-point Likert scale of *'extremely confident'*, *'somewhat confident'*, *'ok'*, *'not very confident'* and *'not at all confident'*.

## Data analysis

Qualitative data were analysed first. The hypothesised program theory was set up in NVivo [41], with theory title as a code, and each context, mechanism (resource and reasoning) and outcome as a sub-code. The interview and then focus group transcripts were coded deductively to the existing sub-codes, and inductively to newly created sub-codes. Analysis progressed with the creation, renaming, merging and deletion of sub-codes, and the coding of interactions between them (context-mechanism; mechanism-outcome)—with the changing structure representing theory refinement. Sample coding was reviewed and verified by the co-authors. The interpretation of data analysis from Arabic speaking women was reviewed and verified by the bicultural researcher.

Quantitative pre and post birth confidence data were analysed for change in birth confidence (as an indicator of change in short-term confidence), using STATA V16 [42]. The pre and post Likert scales were numbered, from '5' for *'extremely confident'* down to '1' for *'not at all confident'*. Chi-square tests were used to assess representativeness of the respondent sample in relation to all women eligible/contacted for both pre and post questions. The Wilcoxon Signed Rank Test was conducted to determine if there was an overall statistically significant difference in the median birth confidence score before and after receiving doula support, and the differential directions of change (increase, no change, and decline in confidence). This test was used because the data were ordinal from Likert scales, comprised two categorical matched pairs (pre-post) [43]. Further Chi-square tests were conducted to determine if there were statistically significant associations between these different birth confidence outcomes and women's sociodemographic, maternal, social/emotional, and program participation factors.

Formal theory, found by literature search, was introduced to assist and deepen explanatory understanding. This involved a process of retroduction—that is, the to-ing and fro-ing between the qualitative data, partly refined program theories, quantitative outcome data and formal theories [44].

This research was conducted with approval from La Trobe University Human Research Ethics Committee (HEC19388).

## Results

### Interview participants

Forty-three English-speaking and nine Arabic-speaking women (total = 52) were eligible during the study period and invited to participate in an interview. Eighteen women expressed interest and were provided with participant information. Thirteen (25% of eligible) provided consent and were interviewed. Participant characteristics are detailed in Table 1.

### Focus group participants

One-hundred and nineteen doulas were invited to participate in focus groups. Half (n = 61, 51%) were active volunteers, 10% (n = 12) were on leave, and 39% were past volunteers. Seventeen doulas (14%) expressed interest and were provided with participant information. Eight

**Table 1. Characteristics of women (clients) interviewed.**

| Characteristic | n |
|---|---|
| Language of interview | |
| English speaking | 7 |
| Arabic speaking | 6 |
| Primary language spoken at home | |
| Arabic | 7 |
| English | 2 |
| Farsi/Persian | 2 |
| Amharic | 1 |
| Tamil | 1 |
| Birthplace | |
| Australia | 1 |
| Outside Australia | 12 |
| Time in Australia | |
| <1 year | 1 |
| 1-5yrs | 5 |
| 6-10yrs | 0 |
| >10yrs | 6 |
| Age | |
| ≤25 | 3 |
| 26–35 | 6 |
| 36+ | 4 |
| Relationship status | |
| Married/partnered | 6* |
| Education level completed | |
| Primary school | 1 |
| Secondary school | 5 |
| Post-secondary diploma/certificate | 5 |
| Tertiary degree or higher | 2 |
| Number of previous births | |
| 0 | 5 |
| 1–2 | 4 |
| 3+ | 4 |

*Relationship status of married/partnered includes two cases of forced international separation due to insecure visa status of partner.

doulas (7%) were available during the study period, consented and participated in one of two focus groups. Participant characteristics are shown in Table 2.

## Routinely collected data of program participants

The program database contained deidentified records of 962 referrals to the program in just over seven years (6 June 2014 to 14 September 2021). The 962 referrals were for 914 individual women as some women were referred and used the program more than once for different pregnancies. The program commenced the routine collection of pre and post birth confidence data from 28 March 2019. Only records of referrals received from this date were considered for analysis (n = 468). Of these, new referrals awaiting intake (n = 20), those referred to another service (n = 30), those who withdrew from the program (125 before being matched

**Table 2. Characteristics of focus group participants (volunteer doulas).**

| Characteristic | n |
|---|---|
| Volunteer status | |
| Currently active | 7 |
| On leave | 1 |
| Past volunteer | 0 |
| Work background | |
| Private doula | 2 |
| Bicultural doula training | 3 |
| Non-practicing midwife/student midwife | 4* |
| Other health/social care | 1 |
| Years volunteering | |
| <1yr | 1 |
| 1-3yrs | 2 |
| >3yrs | 5 |
| Average number of women supported by each doula per year | 3–4 |

*Work background of non-practicing midwife/student midwife includes two doulas with international midwifery qualifications not recognised in Australia.

with a doula and 27 after being matched), inappropriate referrals (n = 5), and those for whom support was still active (n = 38) were removed. Finally, 39 records of women who had not yet been eligible for the post-support follow up questions (because their babies were not yet three months of age), were removed. This left 184 records of women who had been eligible/contacted for pre and post questions. Data for pre and post questions were available for 103 records of women (56% response rate) (Fig 2).

Chi-square tests found the respondent group to be representative of all women eligible/contacted for pre-post questions. No statistically significant differences in sociodemographic, maternal, social/emotional, and program participation characteristics were found between the respondent group and all women routinely eligible/contacted for pre-post questions in the two-and-a-half-year period between March 2019 and September 2021 (see Table 3).

## Refined theory

Analysis of all data sources with formal theories of Recognition by Honneth [45, 46], and Relational reflexivity by Donati and Archer [47] resulted in significant refinement of the program theory. The hypothesised theory was made up of two context-mechanism-outcome (CMO) configurations hypothesising the generation of women's increased confidence. The nature or longevity of confidence was not specified. Refinements to the theory included changes to these initial CMOs, and the addition of a further three CMO configurations—making a total of five CMOs. Three of the five explain differential outcomes in short-term confidence, and the final two explain longer-term outcomes in confidence and related wellbeing. All outcomes are numbered in Fig 3.

**Short-term change.** Outcome 1 represents a short-term increase in confidence felt during the support period. Derived from both quantitative and qualitative data, it is not specific to, but includes an increase in birth confidence. Outcome 2 represents no change in confidence (including birth confidence) felt during the support period and was also found in both quantitative and qualitative data. Outcome 3, found only in the quantitative data, is a decline in birth confidence.

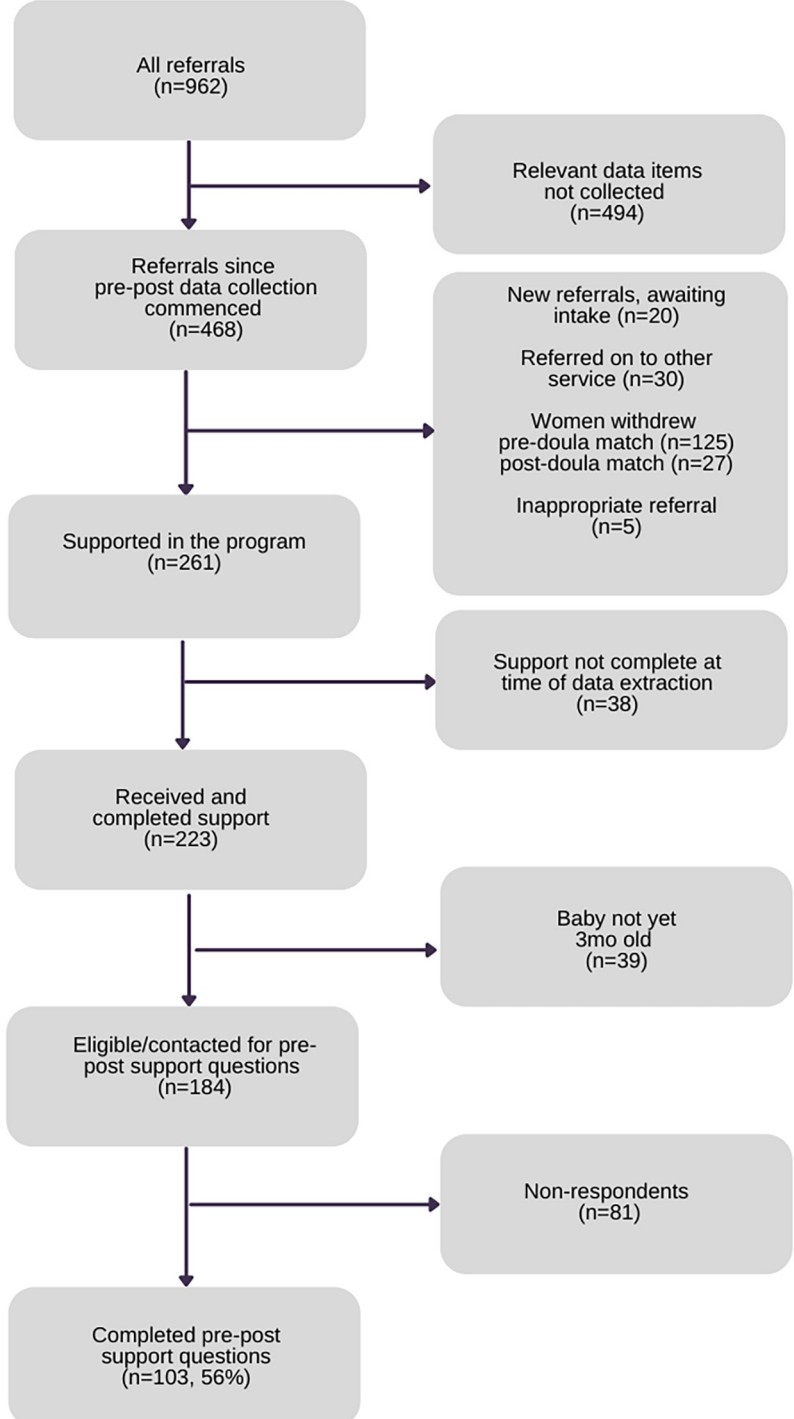

**Fig 2. Routinely collected pre-post birth confidence data available for analysis (respondent group, n = 103).**

Quantitative analysis by Wilcoxon Signed Rank Test of 103 records of women, showed the three birth-confidence outcomes. While the test found a statistically significant change in women's median birth confidence between the pre- and post-time points (z = -2.179, p = 0.03), the direction of change varied. Less than a half of women (n = 45, 44%) had an

**Table 3. Sociodemographic, maternal, social/emotional, and program participation characteristics of respondent group vs all women routinely asked pre and post questions.**

| | Respondents of pre and post birth confidence questions | | All routinely contacted for pre and post questions | | *P-value*[*] |
|---|---|---|---|---|---|
| **Total n** | **n = 103** | **(%)** | **n = 184** | **(%)** | |
| Referral information | | | | | |
| *Socio-demographic* | | | | | |
| Age <25 years | 26 | (25) | 54 | (29) | 0.57 |
| Single | 45 | (44) | 87 | (47) | 0.72 |
| Primary language not English | 61 | (59) | 92 | (50) | 0.41 |
| Interpreter required | 38 | (62) | 60 | (65) | 0.86 |
| Refugee background or seeking asylum | 36 | (35) | 60 | (33) | 0.78 |
| *Maternal* | | | | | |
| First baby | 43 | (42) | 84 | (46) | 0.69 |
| *Social/emotional adversity* | | | | | |
| Experiencing homelessness | 26 | (25) | 49 | (27) | 0.84 |
| History of mental illness | 49 | (48) | 100 | (54) | 0.53 |
| History of abuse or family violence | 52 | (50) | 102 | (55) | 0.66 |
| History of alcohol/other drug addiction | 7 | (7) | 27 | (15) | 0.08 |
| Program participation factors | | | | | |
| Duration of support pre-birth >1mo | 85 | (83) | 140 | (76) | 0.66 |
| Doula attended birth | 70 | (68) | 112 | (61) | 0.57 |

[*]Chi-square test, respondents versus non-respondents.

p<0.05 denotes statistical significance

increase in birth confidence, 28% (n = 29) had no change, and the remaining 28% (n = 29) had a decline in confidence (Table 4).

Potential contextual factors and mechanisms identified in the routinely collected quantitative data were found not to be associated with birth confidence outcomes. Chi-square tests showed no associations between birth confidence and program participation factors (duration of doula support before birth as more than one month, and whether the doula attended the woman's labour and birth). Women's reports of whether their doula helped their birth confidence (yes/no) were also found to have no association with the confidence outcomes. Similarly, women's demographic and other social factors were shown to not be associated with birth confidence outcomes (Table 5). Therefore, these quantitative data did not provide explanatory understanding for differential birth confidence outcomes.

CMO configurations 1–3 assert explanations for outcomes 1–3 (short term change). Each configuration is outlined and evidenced with data. Where formal theories were drawn on, these are also referenced. Codes have been used for qualitative quotations, with D referring to doula, and C referring to client, followed by participant number. CMOs 1–3 are also laid out in Table 6 at the end of this section.

*CMO 1*: *Explaining a short-term increase in confidence*. The outcome of a short-term increase in confidence represents findings from interview data about increased confidence (of any kind) felt during the doula support period, as well as increased birth confidence from routinely collected quantitative data. All interview participants (clients) referred to at least a short-term increase in confidence (or similar concepts), yet quantitative data showed that under half of women actually reported an increase in birth confidence (Table 4). This CMO asserts that when a woman (client) feels misrecognised in her everyday life *(context)*, and trust has been

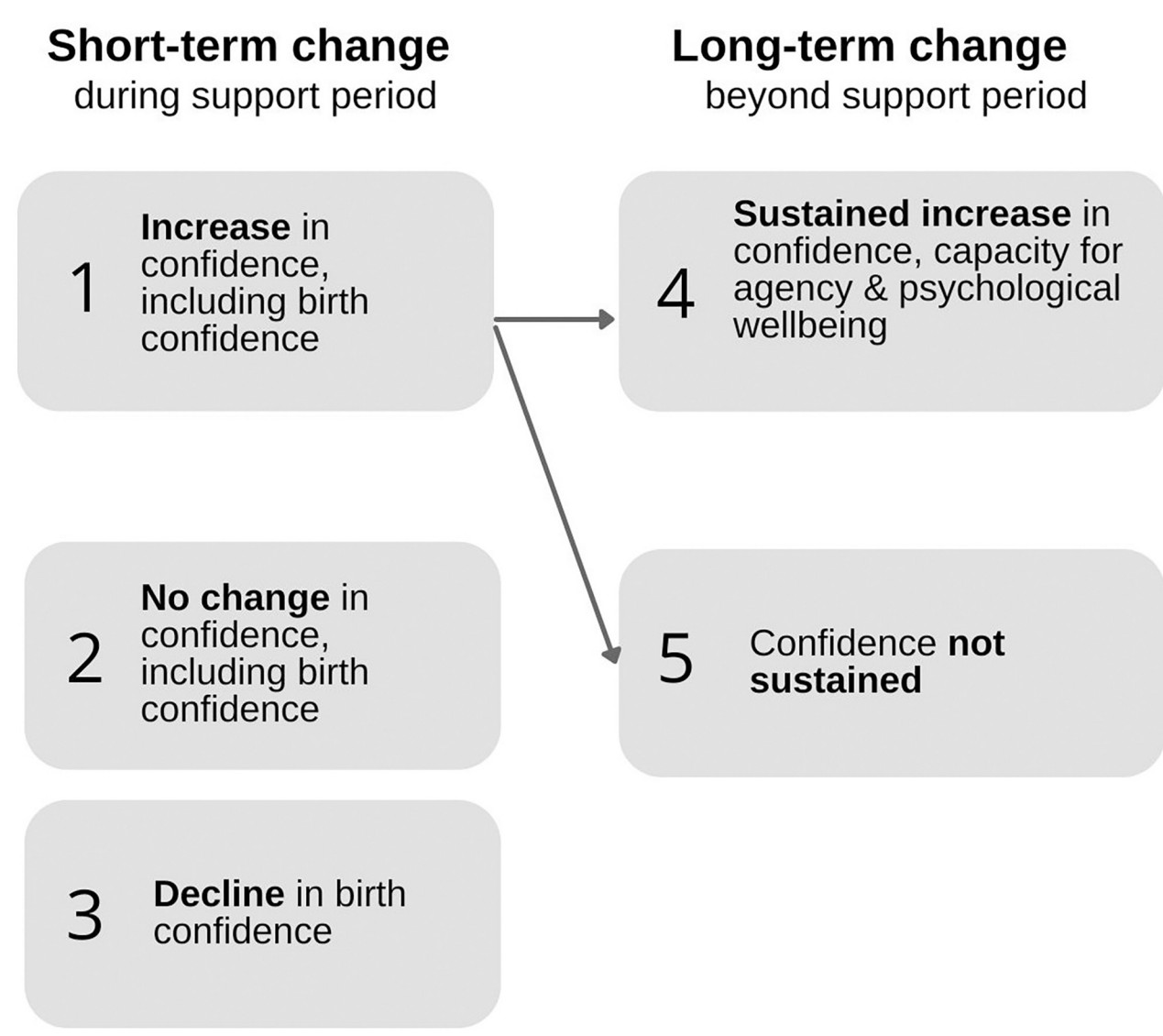

**Fig 3. Short- and long-term confidence-related outcomes for women.**

developed and her story shared with her doula *(context)*, the doula provides recognition of the woman and her life story *(resource)*. In response to recognition, the woman feels less alone in the world, understood and that she can lean on the doula *(reasoning)* which has the outcome of the woman feeling increased confidence during the support period *(outcome)*.

**Table 4. Change in birth confidence outcomes.**

| Outcome | Respondents | | Magnitude of change | |
|---|---|---|---|---|
| | n = 103 | (%) | Pre median* | Post median* |
| Increase in birth confidence | 45 | 44 | 2 | 4 |
| No change in birth confidence | 29 | 28 | 2 | 2 |
| Decline in birth confidence | 29 | 28 | 4 | 2 |

*Out of 5 on Likert scale.

Table 5. Program participation factors and women's social factors' associations with pre-post birth confidence.

| | Increase n = 45 n (%) | | No change n = 29 n (%) | | Decline n = 29 n (%) | | p-value* |
|---|---|---|---|---|---|---|---|
| **Program participation factors** | | | | | | | |
| Doula attended birth | 29 | (64) | 21 | (72) | 20 | (69) | .84 |
| Duration of doula support >1month | 40 | (89) | 22 | (76) | 23 | (79) | .31 |
| Reported that doula helped confidence | 42 | (93) | 29 | (100) | 27 | (93) | .35 |
| **Women's social factors** | | | | | | | |
| First baby | 22 | (49) | 12 | (41) | 9 | (31) | .27 |
| Single/unpartnered | 15 | (33) | 16 | (55) | 14 | (48) | .31 |
| Under 25 years | 10 | (22) | 7 | (24) | 8 | (28) | .87 |
| Primary language not English | 29 | (64) | 14 | (48) | 18 | (62) | .36 |
| Interpreter required | 20 | (69) | 7 | (50) | 11 | (61) | .20 |
| History of abuse or family violence | 19 | (42) | 15 | (52) | 18 | (62) | .25 |
| History of mental health issues | 20 | (44) | 15 | (52) | 14 | (48) | .82 |

*Chi-square test.

P<0.05 denotes statistical significance.

Honneth's theory of Recognition [45] informed the framing of this CMO. 'Recognition' of a woman and her story was 'how' the doula support worked. Recognition is considered a relational resource—a positive and attentive responsiveness to a woman [45, 46]. In this CMO it represents evidence from clients and doulas, of a doula giving time, attention, noticing and working with a woman's strengths and existing resources, positivity, and emotional and physical comfort including touch. It confirms the initially hypothesised resource of 'being by her side', 'getting' her, and believing in her. However, recognition is broader than praising or validating a woman as a mother (initially hypothesised). It is about recognising her personhood [45, 46].

Recognition works in two contexts. First, a woman trusting her doula and disclosing personal information about herself and her life, enables the doula to recognise her. Trust and disclosure are outcomes of a preceding program theory from the same evaluation, asserting that for a woman to trust her doula, the doula needs to show that she is knowledgeable and self-confident. Second, when a woman lives with misrecognition—the denial of dignity and self-esteem in family violence, homelessness, or racism (for example), recognition can powerfully offset this [45, 46].

While quantitative data analyses did not provide explanatory understanding for increased birth confidence, it did rule out both a doula's attendance for labour and birth, and longer

Table 6. CMO configurations 1–3, explaining short-term changes in women's confidence.

| | Context | Mechanism | | *Short-term* Outcome |
|---|---|---|---|---|
| | | Resource | Reasoning | |
| CMO 1 | Woman lives with everyday misrecognition | Doula recognises the woman—her personhood and her life story | Woman feels less alone; feels understood | Woman feels increased confidence while support is active 44% women feel increase in birth confidence |
| | Trust and sharing between doula and woman | | Woman leans on doula for confidence | |
| CMO 2 | Lack of trust and sharing between doula and woman | Lack of recognition (lack of resource) | Woman feels that the doula gave little; there was no benefit | No change in confidence 28% women feel no change in birth confidence |
| CMO 3 | *Unknown* | *Unknown* | *Unknown* | 28% women feel decline in birth confidence |

support duration as necessary contexts for recognition to occur in the generation of birth confidence.

CMO 1 is reflected in the following qualitative data:

*I have a problem with depression and anxiety [and] a bit of problem with my husband as we had a bit of a violent thing. So I felt very lonely. . .my mum is dead, my sister is dead (context), but no, [my doula] is there. Even after I came out of the surgery room, she was still right next to me, just making sure I was okay (resource). . . It was a lot because nobody would [normally] do that (context). . . She gave me so much positivity (resource). . . I thought, with her, I'm going to work through this (reasoning). . . I felt confident with her there (outcome). [The other care providers at the hospital] didn't mistreat me like before. They can't just try to get rid of me, or just rush my appointment, so, yeah. She was very, very confident, and her confidence helped my confidence (reasoning)*

*(C03).*

*Even when it seems like an impossible situation for anyone to be okay (context), we see bravery and, yeah, you find her strengths and skills, and acknowledge (resource)*

*(D10).*

*I was worried about overdoing it with the epidural. I'm worried about overdoing it with any prescribed medication because I'm a recovering addict (context). And she was just really comforting with that stuff and reminding me that it's not heroin, it's an epidural (resource). Yeah, she was just good with that stuff, and she understood. . . We sort of conquered something together, yeah (reasoning). I thought my whole birthing experience would just be like anxiety to the top, but it wasn't (outcome)*

*(C01).*

*Context-mechanism-outcome 2*: *Explaining no short-term change in confidence*. Quantitative data show that 28% of women had no change in birth confidence (Table 4), but only one interview participant reported no change in confidence (of any kind). CMO 2 asserts that in the context of low trust and disclosure (as described in CMO 1) between woman and doula *(context)*, the doula does not or cannot recognise the woman (leaving a *lack of resource)*. In response, the woman reasons that the doula offers little help or benefit *(reasoning)*, and she has no change in confidence *(outcome)*. The following data illustrate this CMO:

*It's easy when I understand the whole story (context). Yeah. When I don't know, then I can't think which way is better to support her. One of my clients wasn't very interested to give me the whole information. . . and I couldn't help her a lot. . .If the connection is very strong (context), then you can support a lot (resource)*

*(D05).*

*We didn't build a relationship (context). She was there but there was not much help (outcome). She didn't touch me or comfort me or anything. She just sat there (lack of resource). I really got nothing from her (reasoning)*

*(C02).*

*Context-mechanism-outcome 3*: *Explaining a short-term decline in confidence*. While 28% of women had a decline in birth confidence evident in the quantitative analysis, no other data

**Table 7. CMO configurations 4 and 5, explaining sustained confidence-related change (or not).**

|  | Context | Mechanism | | *Long-term* Outcome |
|---|---|---|---|---|
|  |  | Resource | Reasoning |  |
| **CMO4** | Woman feels confident during support period (outcome of CMO1) | Doula witnesses and affirms woman's confidence (continued recognition) | Woman sees and feels her own strength and value Woman may also see other people in a new, more positive light. See the good in people. | Woman feels lasting increased confidence, capacity for agency, psychological wellbeing (beyond cessation of doula support) |
|  |  |  |  | *and*<br>Woman may feel strong bond with her new baby. |
|  |  |  |  | She may also feel sorrow and guilt if there are older child/ren with whom she does not have as strong a bond. |
|  |  |  | *and*<br>When support ends, woman reasons she needs to move on, respect the doula, and be grateful for what was had rather than what is gone. | *and*<br>Woman misses doula but copes well when support ends. |
| **CMO5** | Woman feels confident during support period (outcome of CMO1) *(as above)* | Doula witnesses and affirms woman's confidence (continued recognition) *(as above)* | Woman feels support is not enough and is not ready to let go. | Woman does not feel confidence independent of doula or beyond cessation of support. |
|  | *Additional unknown context* |  |  | *and*<br>Woman may feel great loss and distress when support ends. |

collected in this study, or any formal theory, contributed explanatory understanding for this or other short-term decline in confidence. The context and mechanism for this outcome are unknown.

Table 6 outlines all known elements of the generative causation of short-term changes in women's confidence outcomes during the doula support period (CMOs 1–3).

**Longer-term change.** As shown in Fig 3, outcomes 4 and 5 stem from outcome 1, a short-term increase in confidence (via CMO 1). Outcome 4 is a sustained increase in confidence, accompanied by increased capacity for agency and psychological wellbeing, felt beyond the cessation of doula support (at least until the time of being interviewed). A woman may also feel enhanced bonding with her baby. Outcome 5 is when there is no such long-term change. Both outcomes are explained by CMOs 4 and 5. These are evidenced below and described in Table 7.

*CMO 4*: *Explaining a sustained increase in confidence, capacity for agency and psychological wellbeing.* CMO 4 asserts that when a woman (client) feels confident while being supported *(context*, and outcome of CMO 1), and the doula witnesses and affirms the woman's confidence *(resource),* the woman sees her own strength and value. She may also see other people in a new, more positive light *(reasoning).* This results in lasting increased confidence, as well as capacity for agency and psychological wellbeing *(outcome).* In addition, when support ends, the woman reasons that she needs to move on, respect the doula, and be grateful for what was had rather than what is gone *(reasoning).* The woman misses her doula but copes well when support ends (*outcome*). The following data illustrate the CMO:

*I had questions [for the doctor/midwife], and she [doula] would say, "Yes, you can ask, it's your right". And for birth she was present in the room with me, encouraging me (resource). . . and I was doing it! (reasoning). If you're happy during your pregnancy and you feel supported and you're getting enough help and you're confident (context), you continue. It gives you*

*confidence even afterward, and you feel so different (underline:outcome). So as [much as] it hurts, as in like, oh she was such a beautiful person, and helped me so much, understood me, and now I'm not going to see her. . .I'm going to continue [to be] confident*

*(C03).*

*We're validating their personal power. . . the validation, the acknowledgment, being by her. . . (resource). . .we see her capacity to step up into her own thoughts, that which she wants. . . And, yeah, then that empowers her to take that further in her life as a lived experience of, "That worked. I liked it (reasoning). I can see how I might do this again" (outcome)*

*(D07).*

An additional possible outcome in CMO 4 is a woman's enhanced bond with her baby *(outcome)*. However, this may also be accompanied by an unintended negative outcome—feelings of sorrow and guilt for older child/ren with whom she does not have as strong a bond because she was unsupported *(outcome)*. The following interview excerpt illustrates this:

*But with my son, I never had any of this. It was just bang, bang, everything happened. . .no one telling me. . . How miserable I felt and how bad it felt even afterward. . .I was very, very depressed and vague. It affected everything. It affected my relationship with my son because he is not—we're not as tight. . . But with my daughter now it's so different. It is so, so different. She's fully breastfed, and where with my son. . .I felt like it's a crack between me and him and, yeah. But, no, he's my world and I love him so much. . . But there was nobody to actually help me to continue with the breastfeeding, and I was so insecure, and I was so swayed that my milk is not enough, and I'm very sorry about myself with him. I feel guilty about my son that he missed out on this (outcome). What I'm trying to say is that I have two different pregnancies, two different experiences, one with doula, one without, and how much it affects even after the pregnancy*

*(C03).*

*Context-mechanism-outcome 5*: *Explaining no long-term change.* CMO 5 partially explains an absence of long-term change in confidence, agency, or wellbeing. It stems from CMO 4, but asserts that an additional, unknown context results in a woman reasoning differently—that the support is not enough, that she needs more, and she is not ready to let go (reasoning). Any confidence she has felt while supported, falls away when the support ends *(outcome).*

*I really appreciated her walking with me, with the baby in the carrier. . .to sort of just step out-. . .as wasn't comfortable feeding in public. She just helped me pack my bag and she said, "Let's go for a walk, I'm going to walk with you today. You can take your baby out and you'll know that you can" (resource). She kept giving me confidence saying, "It's okay. She's a baby. She's crying, it's fine, but it's new.". . . [But] I felt like I needed a little more because I was still not ready to let go (reasoning). A few more times walking would have been. . . Something to sort of build a little more confidence around taking care of the baby probably would have helped (reasoning)*

*(C06).*

*I wanted her to see me more (reasoning), but unfortunately, she couldn't. I had to sign a contract. I didn't want to message her and get her into trouble, but to be honest, I felt kind of lost (outcome)*

*(C07).*

## Discussion

This paper reports a refined program theory that explains how and when doula support leads to confidence-related outcomes in women. Initial hypotheses, in a theory called *Being by her side*, comprised two pathways to a woman's increased confidence. The first context-mechanism-outcome (CMO) configuration suggested when and how a doula gives time, attention and respect, leading the woman to see her own strength and value as a person. The second was about giving praise, and the woman feeling validated as a mother. Through testing with women (clients) and doulas, and routinely collected client data, the theory was refined and renamed *Recognition*. Its five CMO configurations represent alternative relational-reflexive processes between a doula and woman, leading to five different outcomes. Their key distinctions are about the nature, longevity and and/or absence of a woman's confidence.

The most positive outcome is a woman's short-term increase in confidence felt during the doula support period, that is then sustained beyond cessation of the support (CMO 4 via CMO 1). Increased short-term confidence results from 'leaning on' a doula's confidence. The confidence is not the woman's own (CMO 1). However, when a woman's increased confidence is sustained, it is because the confidence has become her own. It is developed in relationship with, but becomes independent of, the doula. The woman is seeing her *own* strength and value, and when this happens, confidence extends to an increased capacity for agency and psychological wellbeing (CMO 4).

The context that prevents some women with increased short-term confidence from having it sustained, and instead feel loss and distress when the support ends, is unknown. This context affects a woman's reasoning so that she feels the support has not been enough and she is not ready to let go (CMO 5). It is possible that a woman's attachment style could be the explanatory context. According to Attachment theory by Bowlby [48] and its research application by others such as Dempsey [49], a woman with an anxious attachment style may crave emotional care from an attachment figure such as a doula, and find letting go emotionally challenging.

Refinement of the mechanism leading to sustained increase in confidence was supported by formal theories of Recognition by Honneth [45, 46], Relational reflexivity by Donati and Archer [47] and the latter applied to relational care, by Mann [50]. Recognition theory asserts that recognition is an attentive responsiveness to a person that is vital for personal integrity and self-realisation. Relational reflexivity theory suggests that reflexivity—or one's ability to internally balance primary concerns in life with unchosen social circumstances, develops in relationship with others. These theories helped mechanistic understanding—of a doula's care and recognition giving rise to positive changes in the way a woman sees herself, her relationships with others, mothering, and/or other future possibilities. This mechanism also contributes new knowledge about how support during pregnancy, birth and early parenting can stay with a woman long-term. Authors of a recent systematic review of doula and other birth companion care have called for more research into long-term effects [12]. This study contributes that a woman seeing her own strength and value at this time in her life leaves her with more than a positive memory. It can change her future, and if so, her new baby's future. This finding is consistent with another evidence review finding—that maternity care can be healing for women [51], but also builds on this by evidencing how and when this can happen.

The main, more prevalent negative outcomes in the refined theory are no change in short-term confidence (of any kind), or a decline in birth confidence, experienced by over half of women supported in the program. No change is explained by a woman not being able to trust her doula and disclose her concerns and needs in her own words, leaving little for the doula to recognise (CMO 2). It is possible that low trust results from the doula not being woman-centred (as is being tested in another theory in the same study). However, it is also possible that

some women find it hard to trust regardless of a doula's approach. This again may be explained by Attachment theory, but this and/or other new hypotheses would need to be tested.

No explanation was found for a woman's decline in birth confidence (CMO 3; C and M unknown). It is possible that some women 'lose' confidence after not knowing what to expect so start out feeling reasonably confident, but then having this confidence shaken through giving birth (e.g. by traumatic birth), or through participation in the program, and developing a more informed judgement. This judgment could then shape a woman's memory of how she felt at the time of birth. However, it is important to acknowledge that there is a wide range of other contextual factors, such as type of birth and medical intervention, known to impact a woman's retrospective feelings about her birth experience, including confidence [52, 53].

A strength of this study is that it mixed data from multiple sources. Data collection methods also enabled participation of women from diverse cultural backgrounds—albeit only women who could be interviewed in English or Arabic. Results were found to be consistent between English speaking and Arabic speaking interview participants, which suggests methodological rigor and cultural safety for Arabic speakers, or possibly equal levels of socially desirable responding. Other language and cultural groups may reason differently. Social desirability bias may also explain why substantially larger proportions of women said their doulas helped with birth confidence, than the proportions of women whose birth confidence had increased.

Access to routinely collected data was a further strength of the study, including social indicators at the time of referral. We are cognisant with the use of program data that was not collected for research. The social indicators may be underreported (not disclosed) due to shame, stigma and the absence of trust, or variation in understanding/definitions of terms such as trauma, or poor mental health in the absence of a clinical diagnosis [54]. Alternatively, the indicators may be overreported in attempts to strengthen eligibility for the program. The absence of association between women's social factors and change in birth confidence may have been influenced by these issues.

Future iterations of evaluation could build on the findings of this study by asking interview participants for permission to access their routinely collected data. Linking qualitative data on contexts and mechanisms with quantitative outcome data may enable stronger explanations of outcomes (birth confidence changes). Targeted recruitment for qualitative theory testing with women who do not have increased confidence and either continue in the program or withdraw, and doulas who have left the program, may also build on findings. Exploration of new hypotheses including different attachment styles of women as contexts enabling/preventing trust with doulas, mother-baby attachment, and other long-term outcomes, would further refine the program theory. Exploration of the effects of other social supports (eg. friends or family) as contexts affecting how doula support works may be valuable as well. The quality of routinely collected data may also be strengthened by testing and improving the wording of pre and post confidence and other questions. For example, the question, *How confident did you feel about your birth*? could be framed as the baby's birth and be more specific in relation to elements of the birth experience.

The most positive outcome of the program—a sustained increase confidence, capacity for agency and psychological wellbeing from doula support, and knowledge of how and when this occurs, has significant implications for women, the program and for other like programs. The clients of the Birth for Humankind doula support program are from socioeconomic groups known to face significant misrecognition and barriers to a good and healthy life. That the doula support offers a turning point in some women's lives, and they leave the program better emotionally and socially resourced than they were, as women and as mothers, is noteworthy. Knowledge of why and when this outcome occurs may enable it to be maximised, and

alternative negative outcomes minimised through strategic adaptations of the program in the future. This knowledge may also inform other initiatives based on relational care, that seek to improve the confidence and wellbeing of people living with socioeconomic adversity.

## Conclusion

This first realist evaluation of a volunteer doula support program has found how and when a doula's recognition of a woman, in a support relationship, can increase the woman's confidence in the short-term, in a sustained way with broader psychological wellbeing, or not at all.

## Acknowledgments

The authors thank and acknowledge interview and focus group participants, Dr Nawal Abdulghani as bicultural researcher, and Professor Gill Westhorp for methodological guidance.

## Author Contributions

**Conceptualization:** Kerryn O'Rourke.

**Data curation:** Kerryn O'Rourke, Jane Yelland.

**Formal analysis:** Kerryn O'Rourke, Touran Shafiei.

**Funding acquisition:** Kerryn O'Rourke.

**Investigation:** Kerryn O'Rourke.

**Methodology:** Kerryn O'Rourke.

**Project administration:** Kerryn O'Rourke.

**Resources:** Kerryn O'Rourke.

**Supervision:** Jane Yelland, Michelle Newton, Touran Shafiei.

**Validation:** Kerryn O'Rourke.

**Visualization:** Kerryn O'Rourke.

**Writing – original draft:** Kerryn O'Rourke.

**Writing – review & editing:** Kerryn O'Rourke, Jane Yelland, Michelle Newton, Touran Shafiei.

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
