## [Decision Letter · Decision Letter 0]

19 Apr 2022

PONE-D-21-35097How and when doula support increases confidence in women experiencing socioeconomic adversity: findings from a realist evaluation of an Australian volunteer doula programPLOS ONE

Dear Dr. Kerryn O'Rourke,

Thank you for submitting your manuscript to PLOS ONE. After careful consideration, we feel that it has merit but does not fully meet PLOS ONE’s publication criteria as it currently stands. Therefore, we invite you to submit a revised version of the manuscript that addresses the points raised during the review process.

We look forward to receiving your revised manuscript.

Kind regards,

Sharon Mary Brownie

Academic Editor

PLOS ONE

Reviewers' comments:

Reviewer's Responses to Questions

**Comments to the Author**

1. Is the manuscript technically sound, and do the data support the conclusions?

Reviewer #1: Yes

Reviewer #2: Yes

2. Has the statistical analysis been performed appropriately and rigorously? 

Reviewer #1: Yes

Reviewer #2: Yes

3. Have the authors made all data underlying the findings in their manuscript fully available?

Reviewer #1: Yes

Reviewer #2: No

4. Is the manuscript presented in an intelligible fashion and written in standard English?

Reviewer #1: Yes

Reviewer #2: Yes

5. Review Comments to the Author

Reviewer #1: This is a useful article, which can bring a new insight into the care of a new mother of disadvantaged group. The authors have skill fully analysed COM in community based volunteer doula program by realist evaluation. The methodology, data analysis and presentation is done clearly. My recommendation is to publish this article.

Reviewer #2: Thank you for allowing me to review this very interesting paper about how and why doula support to the birth process may strengthen confidence. It is refreshing to read about strategies to effectively change and improve delivery of care to pregnant people.

This is a well written manuscript.

I have only a few comments.

1. pre-post data is referred to in the manuscript, eg line 36 in abstract - I'm curious to know if this pre and post program, or pre and post delivery. At exactly what time points were the pre and post data collected?

2. Introduction, line 49 - I think relational care with continuity could be provided by any practitioner - would just remove (usually a midwife).

3. Line 88, reference is [x] - is a numbered reference missing?

4. Line 90 - since this manuscript is only dealing with. the "being by her side" theory, I'd recommend removing the reference to other theories not evaluated here.

5. Data collection. Did you have a pre-defined set of context, outcomes, or possible mechanisms? Or were these solely. defined by the interviews? Since 'elements of CMO's were put to women' (line 112), I'm wondering if a chart of predetermined contexts, mechanisms, and/or outcomes could be included here. This would help the reader understand exactly what was assessed and then not included in the final models.

6. Did you have any information about other birth support people? (family, friend, partner...). Could other support people in her life be a mediator in these models, or lack thereof possibly explain some of the decline in confidence. I'm wondering if this is important during doula visits as well as at the time of birth. Could other support people pick up on cue's from the doula and hence support or interfere.

6. PLOS authors have the option to publish the peer review history of their article (what does this mean?). If published, this will include your full peer review and any attached files.

Reviewer #1: **Yes: **sanzida Akhter

Reviewer #2: No

---

## [Author Response · Author response to Decision Letter 0]

15 Jun 2022

Please see attached Response to Reviewers letter

---

## [Editor Report · Decision Letter 1]

17 Jun 2022

How and when doula support increases confidence in women experiencing socioeconomic adversity: findings from a realist evaluation of an Australian volunteer doula program

PONE-D-21-35097R1

Dear Kerryn O'Rourke,

We’re pleased to inform you that your manuscript has been judged scientifically suitable for publication and will be formally accepted for publication once it meets all outstanding technical requirements.

Kind regards,

Sharon Mary Brownie

Academic Editor

PLOS ONE

Additional Editor Comments

Reviewer corrects have been addressed.

---

## [Editor Report · Acceptance letter]

23 Jun 2022

PONE-D-21-35097R1 

How and when doula support increases confidence in women experiencing socioeconomic adversity: findings from a realist evaluation of an Australian volunteer doula program 

Dear Dr. O'Rourke:

I'm pleased to inform you that your manuscript has been deemed suitable for publication in PLOS ONE. Congratulations! Your manuscript is now with our production department. 

Kind regards, 

on behalf of

Professor Sharon Mary Brownie 

Academic Editor

PLOS ONE